# AniHead: Efficient and Animatable 3D Head Avatars Generation

## Abstract

Recent advances in diffusion models have led to great progress in generating high-quality 3D shape with textual guidance, especially for 3D head avatars. In spite of the current achievements, the Score Distillation Sampling (SDS) training strategy is too time-consuming for real-time applications. Besides, the implicit representations make these methods generally unavailable in animation. To solve these problems, we present an efficient generalized framework called AniHead, which contains shape and texture modelling respectively for generating animatable 3D head avatars. We propose a novel one-stage shape predicting module driven by parametric FLAME model. As for texture modelling, a conditional diffusion model is finetuned based on the proposed mean texture token. We further introduce a data-free strategy to train our model without collecting large-scale training set. Extensive experiments are conducted to show that our proposed method is not only more efficient than trivial SDS-based methods, but also able to produce high-fidelity and animatable 3D head avatars. The generated assets can be smoothly applied to various downstream tasks such as video and audio based head animation.

## 1 Introduction

Generating 3D content under specific conditions, such as incorporating text, holds significant importance in our 3D world, where we are surrounded by a plethora of 3D objects. The creation of high-quality 3D data through generative models serves as a pivotal enabler for various real-world applications, including virtual reality and cinematic editing. It is worth noting that the generation of 3D data poses unique challenges compared to its 2D counterpart, given the need for intricate and detailed representations. Fortunately, the advent of the Score Distillation Sampling (SDS) technique has emerged as a bridge between 2D and 3D data generation, paving the way for a series of influential works (Poole et al., 2022; Wang et al., 2023b;a) in the domain of text-conditioned 3D generation.

Inspired by the remarkable strides achieved in diffusion-based text-to-3D models (Poole et al., 2022; Wang et al., 2023b), there has been a resurgence of research interest in the generation of 3D head avatars. This endeavor revolves around the creation of highly realistic head avatars, guided by textual descriptions that capture the essence of human characteristics. Existing methodologies (Chen et al., 2023b; Han et al., 2023) typically leverage SDS to independently optimize shape and texture information, building upon pretrained diffusion models. While these SDS-based approaches (Han et al., 2023; Zhang et al., 2023) yield static head avatars of satisfactory quality, they are beset by two inherent limitations that curtail their practical utility: (1) **Excessive Time Consumption**: SDS-based strategies, originally designed for sample-specific optimization, demand the optimization of corresponding objectives for each text prompt through a substantial number of iterations. However, real-world applications often necessitate swift generation to ensure an optimal user experience. Consequently, there is a compelling need for a more generalized generative model that circumvents the need for test-time optimization in 3D head avatar generation. (2) **Challenges in Animation**: Current methodologies predominantly rely on implicit representations, such as DMTet (Shen et al., 2021), for both coarse and refined shape modeling. Regrettably, compared with directly modelling shape through 3D morphable models such as FLAME, implicit representations lack the requisite control over elements like skeletal structure and facial expressions, rendering them ill-suited for animation

purposes. Hence, the realm of 3D head avatar generation stands poised for innovation in the realm of shape modeling, with a pressing demand for alternative strategies to address this shortcoming.

To this end, this paper presents an *innovative, efficient and generalizable* 3D head generation pipeline, aptly named AniHead, that stands as a significant contribution to the field. This pipeline excels in producing high-fidelity, animatable 3D head avatars, and its uniqueness lies in its approach. Our method comprises two distinct conditional generators, each dedicated to modeling shape and texture information with remarkable precision.

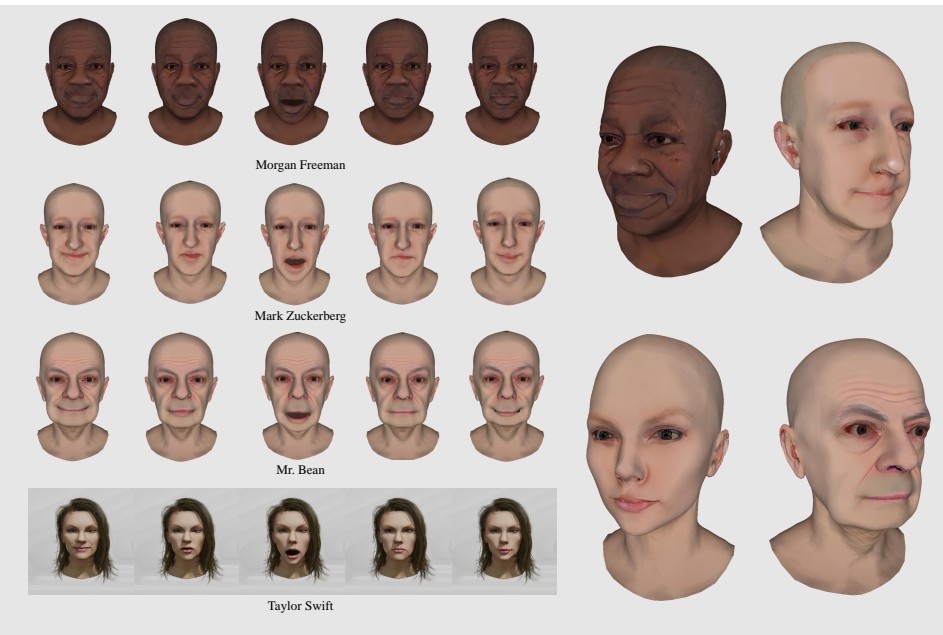

Figure 1: General results of our method with 4K texture. The human head avatars generated by our proposed AniHead can be better adapted in animation, showing mimic expressions. The last row shows the effect of jointly using a downstream CG pipeline as a post-processing step, where selected hair is added to the generated avatar.

Specifically, we enhance the capabilities of the pretrained CLIP text encoder by incorporating an additional MLP module, allowing it to predict control parameters based on textual input. These parameters are subsequently translated to intricate 3D shapes through the use of FLAME, eliminating the need for a cumbersome two-stage generation process as in DreamFace (Zhang et al., 2023) where a coarse geometry is first picked from pre-defined shape candidates according the text prompt and then carved for 300 steps. For the purpose of generalized texture generation, our approach draws inspiration from DreamBooth (Ruiz et al., 2023). Here, we introduce a novel concept of infusing each text prompt with a mean texture token $T^*$ by prepending it to the original prompts. This mean texture token is meticulously crafted to encapsulate essential human head texture information. It can effectively convey the common texture-wise features shared by human beings and UV mapping relation through the finetuning of diffusion model. These enriched prompts serve as the foundation for fine-tuning a pretrained Latent Diffusion Model (LDM), guiding it to acquire high-quality UV maps that seamlessly align with the shape data.

What sets our model apart is its reliance on a data-free training strategy, a stark departure from the resource-intensive data scanning process employed by DreamFace. Leveraging readily available off-the-shelf models (Rombach et al., 2022a; Cherti et al., 2023), we construct a compact training dataset for both shape and texture data through the SDS optimization technique. Once these two generators are trained, our pipeline offers the unique advantage of swiftly generating head avatars tailored to users' requirements, all without the need for additional time-consuming optimization steps. In essence, our contribution revolutionizes the landscape of 3D head avatar generation, offering efficiency, precision, and accessibility in equal measure.

In summary, our paper makes significant contributions to the field of 3D head avatar generation:

1. We introduce an innovative framework, AniHead, for 3D head avatar generation. This novel approach not only produces high-fidelity, animatable 3D head avatars but also excels in terms of inference speed. It achieves this speed advantage by eliminating the need for test-time optimization, ensuring swift and efficient avatar creation tailored to users' requirements.

2. We pioneer a data-free training strategy to empower our generalized generators. This strategy, in stark contrast to labor-intensive manual data creation, leverages readily available off-the-shelf models Rombach et al. (2022a); Cherti et al. (2023) to generate training data efficiently. Our approach not only streamlines the training process but also reduces resource requirements, marking a significant advancement in the field of 3D head avatar generation.

## 2 RELATED WORKS

**Text-to-3D generation**   Text-to-3D models are of great value due to the important visualization and comprehension property of 3D content. However such models were known to be hard to design due to the complexity of 3D data and the correspondence between it and textual input. Fortunately, the recent progress in diffusion-based text-to-image models has inspired various works in the text-to-3D task. Poole et. al. first proposed DreamFusion (Poole et al., 2022), which adapts 2D diffusion models to 3D data. The model is driven by a Score Distillation Sampling (SDS) approach, which can effectively compute gradient through a pretrained diffusion model. Following this method, other works mainly focused on improving performance and efficiency via different designs. For example, Magic3d (Lin et al., 2023) utilizes a two-stage coarse-to-fine strategy to improve the quality. Metzer et al. (2023) proposed to apply SDS to the latent space in order to better facilitate shape guidance. Fantasia3d (Chen et al., 2023b) decouples the original generation into geometry and appearance modelling individually. While our proposed method takes inspiration from DreamFusion to adopt SDS to optimize the shape-related parameters, we design a data-free strategy in which we treat the small scale dataset generated from SDS optimization as ground truth. We then use this dataset to fine-tune the Stable Diffusion model with LoRA strategy. Such a strategy can help our model generate high-quality animatable 3D head avatars within a minute, which can hardly be done by traditional text-to-3d models.

**3D head avatars**   The field of 3D head avatar modelling has garnered increasing attention due to its challenging yet significant nature. Typically previous efforts have been dedicated in generating 3D head avatar from different data sources such as monocular video (Alldieck et al., 2018; Grassal et al., 2022) and text guidance (Han et al., 2023; Zhang et al., 2023). Inspired by text-to-3D diffusion models such as DreamFusion (Poole et al., 2022), several text-based 3D head avatar generation models have been proposed. Concretely, DreamFace (Zhang et al., 2023) proposes to select coarse shape model from ICT-FaceKit through CLIP score matching, which is then refined by optimizing detailed replacement and normal maps in the tanget space. HeadSculpt (Han et al., 2023), on the other hand, adopt FLAME-based NeRF in the coarse stage, and proposes an identity-aware editing score distillation process for refinement. Articulated Diffusion (Bergman et al., 2023) introduces a segmentation loss to enhance the alignment between geometry and texture. Different from these existing methods, we integrate the parametric FLAME model for smoothly animating the generated head avatars. Moreover, we utilize a generalized texture model instead of identity-specified one, thus resulting in more efficiency.

**Personalized control for diffusion models**   While the pretrained diffusion models can learn large amount of prior knowledge from training data to construct high-quality data, it has been a problem to drive these models with some personalized requirement, e.g. generating images of a specific species. Textual Inversion (Gal et al., 2022) and DreamBooth (Ruiz et al., 2023) provides two different solutions. Concretely, Textual Inversion proposes to optimize an additional word, while DreamBooth chooses to finetune the pretrained diffusion models with pre-defined identity token. Other works try to apply these methods to different tasks. For example, DreamBooth3D (Raj et al., 2023) extends DreamBooth to 3D generation; AnimateDiff (Guo et al., 2023) attaches motion-related parameters to subject-driven image generation model for animation; DisenBooth (Chen et al., 2023a) introduces contrastive objectives to improve the information represented by the identity token. Our proposed method also utilizes a similar technique to replenish our text prompts. The difference between the existing methods and ours lies in the specific application domain. While the identifier in previous

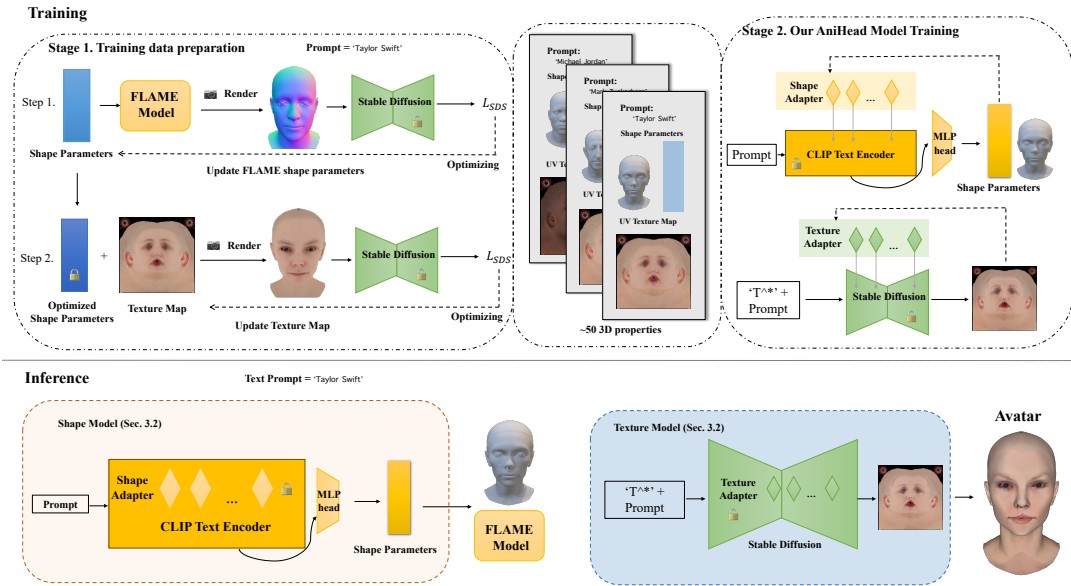

Figure 2: Schematic illustration of our proposed pipeline. We first adopt SDS technique to create training data. Then the generated training set is used to train both the shape and texture generators. The text prompts are projected to FLAME parameters by the shape generator and UV maps by the texture generator with the assistance of mean texture token. During inference, our model can directly generate high-quality human head avatars without test-time optimization.

methods mainly stands for subject-specific characteristics, we adopt such a method to guide the diffusion model with mean facial features among human beings.

## 3 METHODOLOGY

In this section, we present our novel method for generating head avatars as shown in Fig. 2. We build our model based on the observation that 2D priors embedded in the foundation models can be effectively elevated to 3D representations. We propose to generate head avatars through a two-branch network (Sec. 3.2) consisting of shape generator and texture generator. Specifically, these two modules predicts shape and texture data respectively corresponding to the text prompts, by adapting representative pretrained models such as CLIP and LDM to the new data domain. By fusing these two components, we obtain a unified model capable of generating high-quality head avatars with remarkable efficiency. We further employ the SDS loss to extract shapes and textures aligned with given source prompts (Sec. 3.3), which are then used as training data for our model.

### 3.1 PRELIMINARY

**Score Distillation Sampling** The Score Distillation Sampling (SDS) technique proposed in DreamFusion (Poole et al., 2022) has largely facilitated the design of text-to-3D models. This method utilizes a pretrained text-to-image model $\phi$ such as Imagen (Saharia et al., 2022) and Stable Diffusion (Rombach et al., 2022b) to optimize a neural radiance field (NeRF) parameterized by $\theta$ through enhancing its consistency with textual information. The key insight of SDS is to generate 3D models that produce visually good images when rendered from various random viewpoints. In other words, SDS elevates 2D representations to 3D by ensuring that the differentiable rendering results of 3D objects closely match the powerful text-to-image model, such as Stable Diffusion, to obtain the shape and color of 3D objects.

Specifically, given an input sample $x$, corresponding latent feature map $z$ and its noisy version $z_t$, where $t$ denotes the time step, and the corresponding textual information $y$, SDS first estimates the noise added to $z_t$ as $\epsilon_\phi(z_t; y, t)$. This estimation is then transformed into the loss function of

difference between predicted and added noise, as following:

$$\nabla_\theta \mathcal{L}_{\text{SDS}}(\phi, x) = \mathbb{E}_{t,\epsilon} \left[ w(t)(\hat{\epsilon}_\phi(z_t; y, t) - \epsilon) \frac{\partial z}{\partial x} \frac{\partial x}{\partial \theta} \right], \tag{1}$$

where $\epsilon$ is the added random noise, $w(t)$ controls the weight of loss based on $t$, $x$ being the differentiable rendered image of the 3D object. In the SDS method, Classifier-Free Guidance (CFG) (Ho & Salimans, 2022) introduces a guidance scale parameter $w_e$ is introduced to improve the sample fidelity while balancing the diversity of the generated samples. Denote the guided version of the noise prediction as $(\hat{\epsilon}_\phi)$. The guidance scale parameter $w_e$ is incorporated as follows:

$$\hat{\epsilon}_\phi(z_t; y, t) = (1 + w_e)\epsilon_\phi(z_t; y, t) - w_e\epsilon_\phi(z_t; t). \tag{2}$$

where $\epsilon_\phi$ is the pre-trained denoising function. The guidance scale parameter $w_e$ adjusts the score function to favor regions where the ratio of the conditional density to the unconditional density is high. In previous works such as DreamFusion, Fantasia3D, and HeadSculpt, they tend to set the guidance scale parameter $w_e$ to a higher value to enhance text control capabilities, enhancing sample fidelity at the expense of diversity. However, since we employ the strong prior knowledge provided by the FLAME model, we set this parameter to a relatively low value and obtain more realistic, real-life outcomes. In this paper, we mainly utilize this technique to generate training data for our propose generalized shape and texture generators.

**Parametric FLAME model**  FLAME (Li et al., 2017) is a 3D morphable model (3DMM) for human faces that combines shape, expression, and pose variations. As a parametric human head model capable of processing corresponding parameters into human head meshes, FLAME employs Principal Component Analysis (PCA) to derive compact representations for various facial components, including shape and expression variations. Specifically, the model is defined as

$$T_P(\boldsymbol{\beta}, \boldsymbol{\theta}, \boldsymbol{\psi}) = \mathbf{T} + B_S(\boldsymbol{\beta}; \mathcal{S}) + B_P(\boldsymbol{\theta}; \mathcal{P}) + B_E(\boldsymbol{\psi}; \mathcal{E}) \tag{3}$$

$$M(\boldsymbol{\beta}, \boldsymbol{\theta}, \boldsymbol{\psi}) = W(T_P(\boldsymbol{\beta}, \boldsymbol{\theta}, \boldsymbol{\psi}), \mathbf{J}(\boldsymbol{\beta}), \boldsymbol{\theta}, \mathcal{W}), \tag{4}$$

where $\boldsymbol{\beta}$ denotes shape components, $\boldsymbol{\theta}$ denotes pose with regard to neck, jaw and eyeballs, $\boldsymbol{\psi}$ denotes expression, $\mathcal{S}, \mathcal{P}, \mathcal{E}$ denote space for previous three attributes, $W$ is a blend skinning function. Compared with the implicit representations such as DMTet, FLAME model can better control the detail of head shape, thus being more appropriate for animation. In addition to shape and pose components, the FLAME model also predefines a skeletal structure, which all facilitate driving and customizing expressions and actions within computer graphics (CG) pipelines.

## 3.2  Model Structure

**Shape Model**  While SDS primarily employs implicit representations, such as DMTet and NeRF, to generate a wide range of objects, the task of generating human head avatars has a relatively narrower scope but demands a higher level of detail. Therefore, it requires a more constrained yet fine-grained approach. Furthermore, the generation of avatars is intended for downstream applications, such as talking head animations. It is thus crucial to use parametric models that can adapt to computer graphics pipelines, rather than relying on implicit shape representations. To this end, we utilize the FLAME parametric model as our shape representation, which possesses fundamental 3D human head structures and can easily control the head shape.

Formally, the shape data is parameterized based on FLAME, denoted as $S_i \in \mathbb{R}^m$, where $m$ is the dimension of PCA shape parameter. We propose to learn a generalized shape generator $F_S$ that predicts $S_i$ conditioned on arbitrary text prompt $T$. In particular, we leverage the pretrained CLIP text encoder to extract the text embedding $t = \text{CLIP}(T)$. $t$ is then projected to the same parameter space as $S$ via an additional MLP module $f_\theta$ to produce the predicted shape parameter $\hat{S} = f_\theta(\text{CLIP}(T))$. With post-processing steps that include the addition of accessories such as hats and hair, the generated avatars can be readily employed in downstream tasks. Note that there are actually lots of pretrained text encoders, such as T5. And it has been found that using such powerful models can significantly improve the quality of generation in diffusion models (Balaji et al., 2022). Different from most text-to-image and text-to-3D models that utilize fixed text encoders, we try to help the model learn shape-aware knowledge through finetuning. Therefore the CLIP text encoder is chosen for simplicity.

**Texture Model**   For texture modelling, we seek a straightforward way to finetune a pretrained Latent Diffusion Model (LDM) so that we can make use of the generalizable prior knowledge in such a model. Formally, the LDM is further trained to generate UV maps based on textual guidance. However, UV maps generally have remarkable domain gap with normal images. Besides, text prompts can only provide those significant characteristics of a person, e.g. gender, age, etc. In contrast, the most basic prior knowledge of human head texture, such as the topology of human face, UV mapping relation, common facial features, etc., is often missed in the prompts. To overcome this issue, we propose to complement each the prompts with a pre-defined mean-texture token $T^*$. Specifically, we generate a modified version $\tilde{T}_i$ for each text prompt $T_i$ by prepending $T^*$ to it. The mean-texture token is aimed to represent prior knowledge of human face textures. By this means, the renewed text prompts can have sufficient content so as to denote the general human facial features, other than the original prompts that often point to specific features of each human face. By finetuning the LDM with $\tilde{T}_i$, the model can gradually learn the expected information contained in the mean texture token, thus improving the generaion quality. It is noteworthy that while the proposed mean-texture token is similar to the unique subject identifier utilized in DreamBooth, the actual effect is indeed different. The identifier in DreamBooth is aimed to represent a specific subject. In contrast, our mean-texture token helps complement text prompts by representing common texture-wise features shared by human beings.

## 3.3   TRAINING STRATEGY

**Shape Model**   Given the promising capabilities of SDS-based text-to-3D methods, we propose to build the training pipeline based on such methods. We first prepare a few candidate prompts $\mathcal{T} = \{T_i\}_{i=1}^n$ ranging from descriptions of celebrities to specific portrayals of individuals. This endows our shape model with the generalization ability to generate shapes that both reflect identity and conform to descriptions. Then by applying the SDS loss as in Eq. 1, we optimize the shape parameters $S_i$ corresponding to each prompt $T_i$, which results in the pairwise text and shape data. Afterwards these data is utilized to train our proposed shape generator, in which the parameters of MLP $f_\theta$ are optimized along with those of the CLIP text encoder. To keep the efficiency, the text encoder is finetuned in the LoRA style, i.e. learning newly-introduced parameters, that are low-rank decomposed, for each layer.

**Texture Model**   We adopt the same training strategy as for the shape model, i.e. generating training UV maps via SDS optimization based on texture pool $\mathcal{T}$ and finetuning the pretrained LDM. As introduced in Sec. 3.1, the FLAME parametric model is implemented using PCA. However, the texture component is subject to dimensionality constraints after PCA, leading to an inability to express high-frequency details and causing the model to suffer from limited texture representation capabilities. Therefore, we only utilize the texture map UV relationships from the FLAME model and apply the texture UV map $\Psi \in \mathbb{R}^{H \times W \times 3}$ to the head shape obtained in Sec. 3.2. Specifically, we represent the texture as $\Psi = \bar{\Psi} + \Delta_\Psi$, where $\bar{\Psi}$ denotes the mean texture provided by the FLAME model and $\Delta_\Psi$ denotes human-specific texture details. $\Delta_\Psi$ enables more refined optimization results, i.e. personalizing each texture given the corresponding text prompt. Therefore, we optimize the 0-initialized $\Delta_\Psi$ using SDS loss to produce high-fidelity textures with strong identity. As for general model, to achieve both training efficiency and generalization capabilities, we fine-tuned Stable Diffusion using the LoRA style, which is consistent with the approach used for the Shape Model.

# 4   EXPERIMENTS

## 4.1   IMPLEMENTATION DETAILS

Generally, the FLAME parametric model has a total of 300 shape parameters processed with PCA. We choose to optimize the first 100 dimensions, which are sufficient to express facial shapes, to avoid artifacts such as excessively angular facial shapes. The parameters are clamped within the range [-2, 2] for better results. The size of UV maps are set as the $256 \times 256$ and the resolution is increased to 4K during inference using off-the-shelf methods. Specifically, our $256 \times 256$ textures are first restored to $1024 \times 1024$ using RestoreFormer++ (Wang et al., 2023c), then to 4K using superresolution model Real-ESRGAN (Wang et al., 2021).

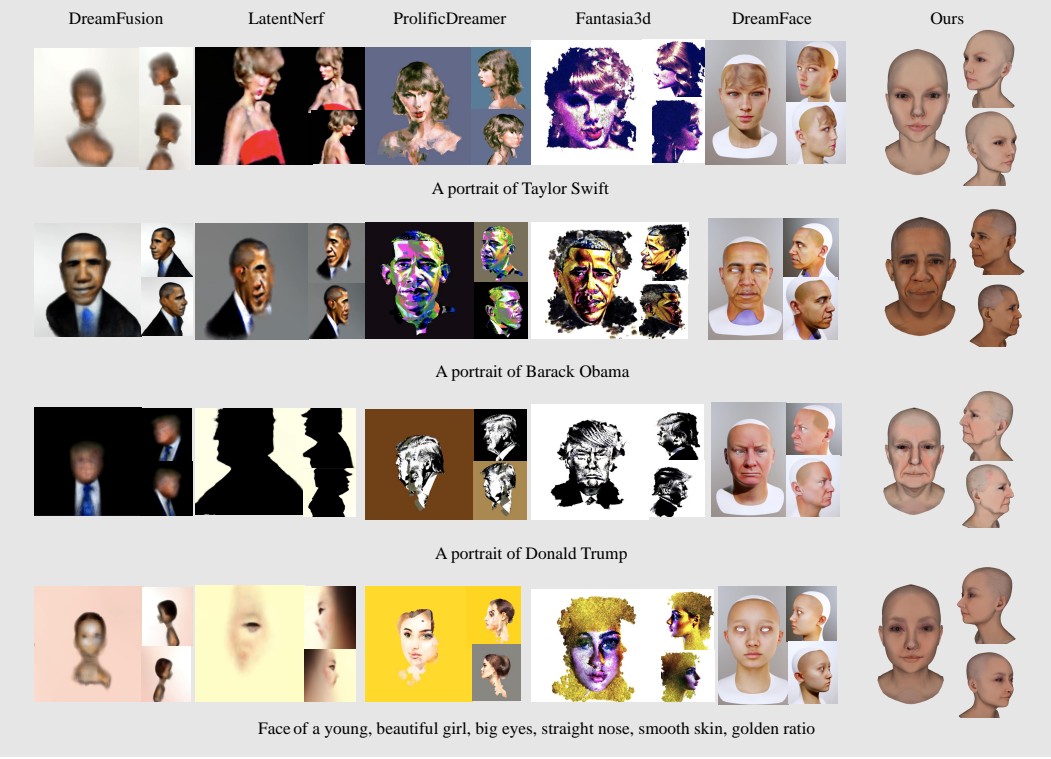

Figure 3: Comparison with existing text-to-3D methods. While other methods may fail to generate reasonable outcomes, our approach stably produces high-quality shape and texture, leading to superior results within less time.

We obtained a dataset of over 600 samples through SDS optimization. We then picked 50 samples out of them, ensuring a balance of gender, ethnicity, and age, including both celebrities and descriptive text results. This selection process effectively increased the generalization capabilities of our AniHead models. For the CLIP text encoder, we apply the LoRA strategy to the Query and Value (QV) layers of the CLIP text encoder. We use learning rate initialized as $1e-3$, linear decay to 0, batch size being 128 and AdamW optimizer, with attention dimension $r$ 16 and adaptation coefficient 16. For Texture Model, we use learning rate initialized as $1e-4$ and cosine decay, with 10 step warmup, AdamW optimizer, attention dimension $r$ 128 and adaptation coefficient 64. The other details are listed in the appendix.

## 4.2 QUALITATIVE RESULTS

We present our qualitative results in two main parts, i.e. static human heads and animated ones. For the static results, we present comparison with other text-to-3D methods including DreamFusion (Poole et al., 2022), LatentNerf (Metzer et al., 2023), ProlificDreamer (Wang et al., 2023b), Fantasia3d (Chen et al., 2023b) and DreamFace (Zhang et al., 2023). The results are shown in Fig. 3. DreamFace suffers from limited prior shape information since it requires selecting coarse shape models from ICT-FaceKit, which only contains 100 identity base models. As a result, this method struggles to generate shapes that closely match the prompts, leading to final results with weak identity features while our model is able to generate 3D models bearing distinguish identity. On the other hand, although the optimization-based approaches such as LatentNeRF and ProlificDreamer can have better quality than DreamFusion, the results stem from 2-3 times repetition and selection. This is because they have unstable optimization directions and are prone to producing failure cases. Besides, We also observed the "Janus problem" in ProlificDreamer and LatentNeRF (as seen in Row 3 and Row 4) even when using the Perp-Neg strategy (Armandpour et al., 2023). In contrast, our method, which is guaranteed by the FLAME model prior, can stably generate realistic shapes and textures that accurately reflect identity features.

Beyond creating head avatars with general prompts, our method can also well handle specific prompts for tasks such as generating special character and editing. Although we only train our model on celebrities and real-life people, the shape and texture generators can generalize well to special characters such as Batman thanks to the strong prior provided by Stable Diffusion. Furthermore, our model enables editing directly through natural language instructions. For example, in Fig. 4, we achieve age and style variations by simply adding "old", "as a clown" or "white/brown skin" to the prompt.

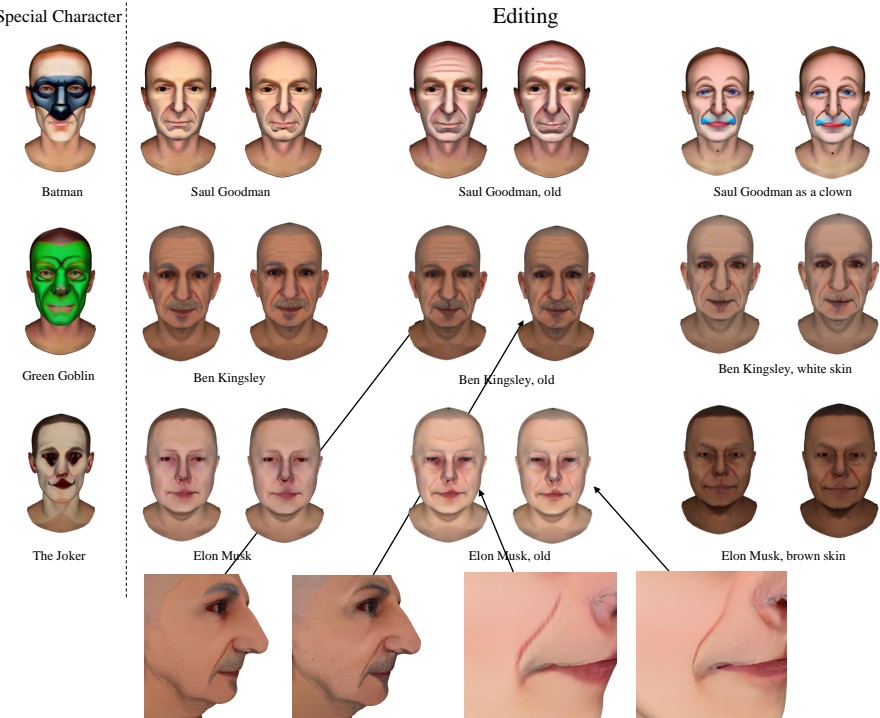

Figure 4: Generate special character and editing through natural language. Avatars of $256 \times 256$ resolution are shown in the left part, with 4K resolution results in the right part. By zooming in, we can see that details such as eyebrows, beards, spots, and wrinkles are all clearer and more realistic.

As for the animation, thanks to the convenient control of FLAME model, we can drive our generated avatars using pretrained DECA (Feng et al., 2021) model, of which the results are presented in Fig. 5. Specifically, we adopt DECA to extract expression and pose parameters, and then merge them into our generated avatar, thereby enabling the animation of the avatar. As we can see in Fig. 5, by directly using a simple off-the-shelf FLAME parameter extraction model, we can achieve video-based driving, with the resulting animation being continuous and natural. It is noteworthy that the resulted heads suffer from drawbacks like unnatural mouth and eye movements. These problems can be ascribed to DECA's lack of specific training for the eyes and insufficient focus on the mouth region during training. We have the potential to achieve better results with an improved FLAME parameter extraction model, or by utilizing motion capture data directly to drive the FLAME model for more accurate and natural animations.

### 4.3 QUANTITATIVE RESULTS

For quantitative results, we mainly focus on two important metrics. The first is CLIP score (Radford et al., 2021), which measures the fidelity of generated results. Higher CLIP score indicates better quality. We borrow the results from DreamFace and follow its implementation by calculating the cosine similarity between image features and text features, using the prompt "the realistic face of NAME/description". The second is the inference time for each sample, which can sufficiently show the efficiency of each method. As shown in Tab. 1, our method demonstrates realism, good text alignment, as well as efficiency, outperforming other methods in terms of both CLIP score and running time.

Figure 5: Animation results of our proposed method. We drive the generated avatars using DECA, with a video of Barack Obama as source video.

| Method | CLIP score ↑ | Inference time ↓ |
|---|---|---|
| Text2Mesh | 0.2109 | ∼15 mins |
| AvatarCLIP | 0.2812 | ∼5 hours |
| Stable-DreamFusion | 0.2594 | ∼ 2.5 hours |
| DreamFace | 0.2934 | ∼ 5 mins |
| **Ours** | **0.3161** | **∼ 1min** |

Table 1: Comparison of CLIP score and inference time for one sample between our proposed method and the previous ones.

## 4.4 MODEL ANALYSIS

**Effectiveness of mean texture token.** To further show the efficacy of the proposed mean texture token, we conduct an ablation study among models utilizing two variants of mean texture token as follows: (1) **Rare Token**: the method proposed as in Sec. 3.2, and (2) **Natural Language Description Token**: instead of using a special token, we prepend a phrase "Facial UV texture" to the original prompts. As shown in Fig. 6, using only the mean texture token and gender-specific words such as "man" and "woman" can yield mean textures, even though the mean texture image was not included in the training dataset. On the contrary, the Natural Language Description Token cannot generalize to special cases, such as special characters and stylized editing. This may be because the Natural Language Description Token contains and reinforces the real-face prior, while the Rare Token can better learn the UV texture mapping.

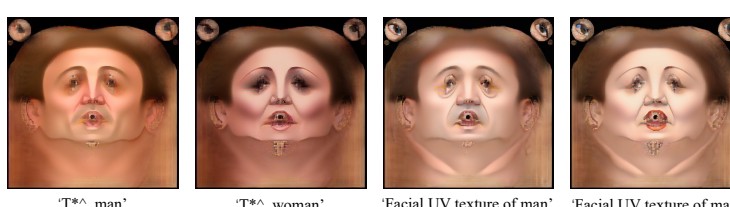

'T*^, man'          'T*^, woman'          'Facial UV texture of man'          'Facial UV texture of man'

Figure 6: Generated Texture from mean texture tokens. During training, we did not include the FLAME mean texture image for fine-tuning. However, the model can still learn to associate the mean texture token with the mean texture map.

## 5 CONCLUSION

In this paper we discuss the merits and drawbacks of the current SDS-based text-to-human head models. Based on the discussion we explore a new pipeline which adopts SDS technique to generate training data for generalized model rather than directly using it in inference. The proposed pipeline learns a shape generator driven by FLAME-based parameters and a texture generator which utilizes text prompts complemented by our proposed mean texture token for generations. The extensive experiment results show both the efficiency and effectiveness of our method, indicating the strength of such pipeline when generating high-quality animatable human heads in real-life applications.

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

## A  APPENDIX

### A.1  ADDITIONAL IMPLEMENTATION DETAILS

**Camera angle range**  Unlike other SDS-based methods, since we optimize on a reasonably good prior human face, we do not require a large camera angle range. In our experiments, we found that excessively large elevation angles can also introduce artifacts, possibly due to the discrepancy between non-daily-life camera angle images and the Stable Diffusion prior.

|  | Shape Optimization | Texture Optimization |
|---|---|---|
| Elevation Range | $[-5, 5]$ | $[-5, 5]$ |
| Azimuth Range | $[-90, 90]$ | $[-90, 90]$ |
| CFG | 50 | 7.5 |
| Learning Rate (lr) | 0.001 | 0.0008 |
| Optimizer | AdamW | AdamW |
| Weight Decay | 0.01 | 0.1 |

Table 2: Hyperparameter configuration during SDS optimization.

|  | Shape Model | Texture Model |
|---|---|---|
| Learning Rate (lr) | 0.001 | 0.0001 |
| Learning Rate Decay Strategy | Linear | Cosine |
| Batch Size | 128 | 8 |
| Optimizer | AdamW | AdamW |

Table 3: Hyperparameter configuration during generalized model training.

**Initial pose**  Additionally, the initial pose of FLAME has a slight upward tilt. We observed that, during experimentation, excessively large camera azimuth angles (such as a 90° side view) could also yield poor results due to the difference from the Stable Diffusion prior. Therefore, we crawled 100 ID photos from the internet and used DECA (Feng et al., 2021) to detect the average FLAME pose in these photos. We fixed FLAME model on this pose and then optimized the shape parameters.

**SDS Training Details**  Since we want to generate decoupled shape and texture instead of single 3D shape, we have to optimize two sets of parameters controling shape and texture respectively. Specifically, shape paramter $S \in R^{1 \times 100}$ is optimized with SDS. $S$ can be mapped to 3D head shape by fixing the pose and expression parameters to 0 in the FLAME model. Formally, $T, M_{normal} = FLAME(S)$, where $T$ is mesh shape and $M_{normal}$ is the according normal map. Then, we use a differentiable renderer $R$ with a random camera pose $C$ to obtain the shader image $I = R(T, C, M_{normal})$. We then encode $I$ into the latent space to obtain $z_t$, and we use the SDS loss to get the optimized shape $S'$.

For texture, we fix the shape parameters $\hat{S} = S'$ and set the parameter to be optimized as texture map $M_{\text{tex}}$, which is initialized as the mean texture. Similarly, we obtain $I = R(T, C, M_{\text{tex}})$, which reflects the effect of mapping the texture map to be optimized onto the 3D shape.

**Weight Decay**  In our experiments, we discovered that, owing to the instability of the SDS loss, it is necessary to apply a larger weight decay. This is combined with the texture strategy, defined as texture $\Psi = \bar{\Psi} + \Delta_\Psi$, where $\bar{\Psi}$ denotes the mean texture provided by the FLAME model and $\Delta_\Psi$ denotes human-specific texture details, and we optimize only $\Delta_\Psi$. Compared to optimizing the entire texture map, focusing on optimizing $\Delta_\Psi$ and increasing weight decay leads to more realistic texture maps. In contrast, optimizing the entire texture map may result in the introduction of unwanted artifacts, such as points and lines.

**Classifier-free guidance**  Furthermore, former SDS-based methods, such as DreamFusion (Poole et al., 2022), tend to use high classifier-free guidance (CFG) (Ho & Salimans, 2022) to improve sample fidelity at the cost of introducing artifacts and causing generated colors to be overly smooth and distorted. Since our method already incorporates a strong prior, we do not need a large CFG to achieve good optimization results. In fact, we found that with a CFG of 30, the generated results are not sufficiently realistic, resembling stylized paintings. A CFG below 5 leads to overly blurry results, indicating insufficient text control. By setting the CFG to 7.5, following the conventional value for Stable Diffusion, we can generate realistic, high-fidelity results with great identity matching. The relevant results are shown in Fig. 7.

**Discussion about FLAME model**  Admittedly, FLAME may have limitations in representing certain accessories such as hair or hats. However, using a parametric model has its advantages:

CFG value

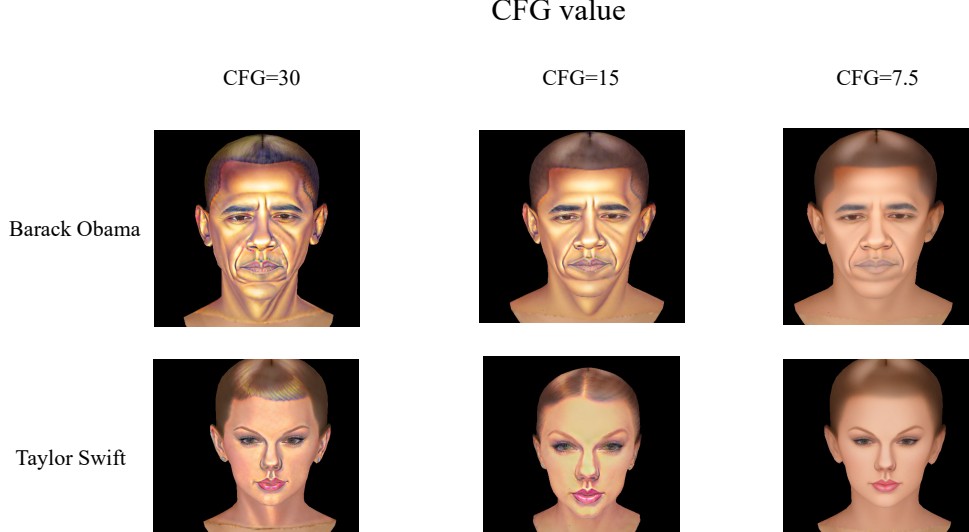

Figure 7: The effect of various CFG values. We finally set the CFG to 7.5 for better fidelity.

(1) The parametric model incorporates a human head prior, providing constraints that enable the generation of more reasonable results, while implicit representation methods struggle or fail to generate reasonable results (as shown in Fig.3). Both shape and texture are more controllable and reasonable, which had also mentioned in our overall response.

(2) Our approach is animatable, allowing for easy integration with off-the-shelf driving methods to create animations, as demonstrated in the gif in Figure 5. In contrast, implicit methods, although capable of representing hair, are not animatable and lack downstream applications.

(3) For static 3D head avatars, adding hairs by post-processing, as adopted by us and DreamFace Zhang et al. (2023), would not lead to significant performance gap with latent-based methods.

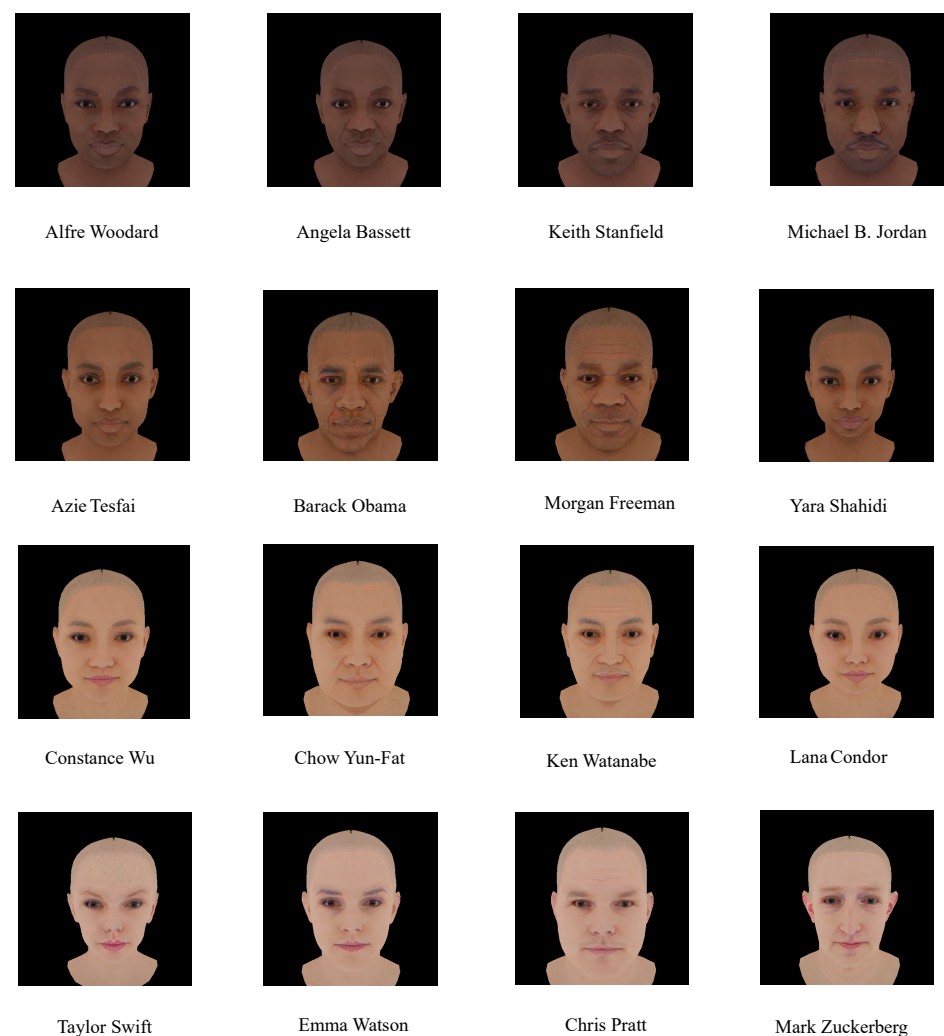

Figure 8: Celebrities generated by our method.

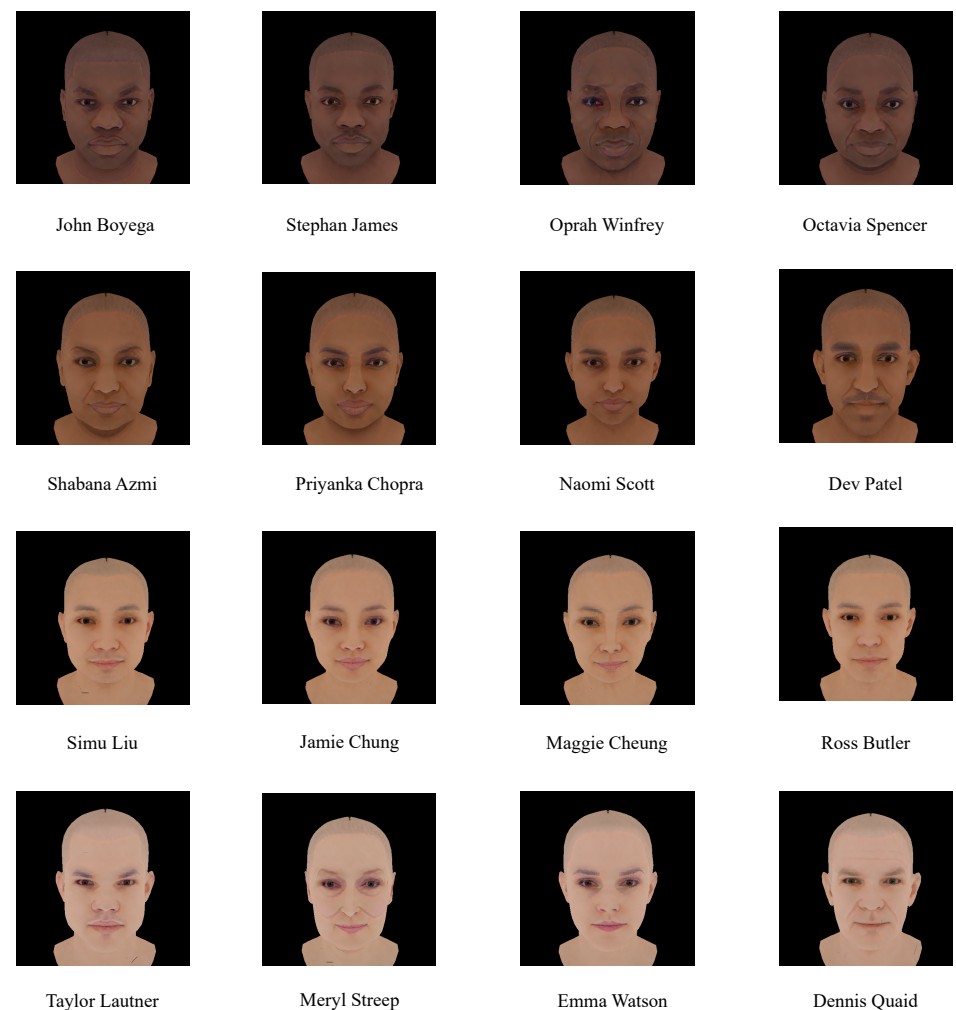

Figure 9: Celebrities generated by our method.

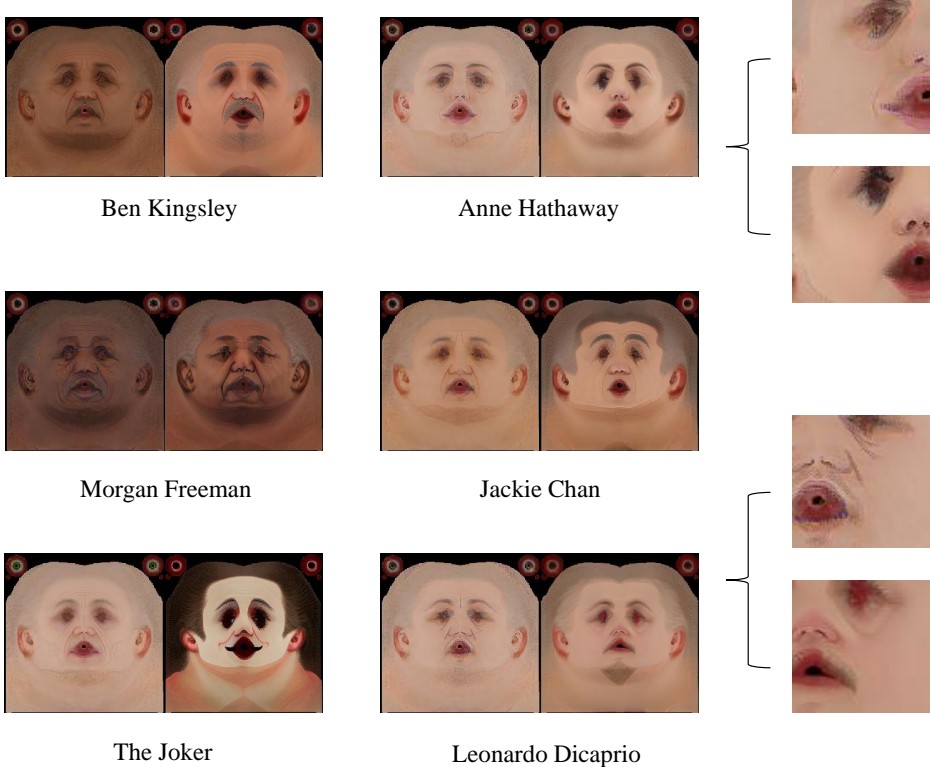

Ben Kingsley

Anne Hathaway

Morgan Freeman

Jackie Chan

The Joker

Leonardo Dicaprio

Figure 10: We show part of the SDS optimization results (left of each set) and AniHead generation results (right of each set). Specifically, we compare the SDS optimization results with the inference results and show that not only does inference results generates better in skin colors, but also better in details. We zoom in two sets of results to show improvement in details.

Prompt = 'Elon Musk, brown skin'

w/o. $T^*$      w. $T^*$

Prompt = 'girl'

w/o. $T^*$      w. $T^*$

Figure 11: Comparison of generation results with and without the proposed mean texture token. We can find results without mean texture token $T^*$ suffer from low alignment with the text prompts. Meanwhile, the generated textures have problems of spatial misalignment. We've added guides (orange lines) to illustrate that without $T^*$, the the UV geometry of delicate parts may be misaligned, such as the nose and eye regions. We've also zoomed in on these two areas from generated 3D heads to show the error caused by the texture UV position offset.

