# OpenReview forum: "AniHead: Efficient and Animatable 3D Head Avatars Generation"
_ICLR.cc/2024/Conference — Submitted to ICLR 2024_

### Official Review · Reviewer_3kTT · 2023-10-29

**Soundness:** 2 fair
**Presentation:** 3 good
**Contribution:** 2 fair
**Rating:** 5
**Confidence:** 4

**Summary:**

This paper studies the problem of text-guided 3D face generation and proposes a method to achieve 3D head generation with one feed-forward pass without test-time optimization.  Here 3D parametric head model and texture maps are used to represent 3D heads. This work first generates texture maps and shape parameters based on text prompts by optimizing the 3D head parameters and texture maps using standard SDS loss. In this way, a set of samples with text and corresponding 3D heads are generated. These samples are then used to train models to directly predict the 3D head representation from text. Here, the shape parameters are predicted by an MLP with CLIP text embedding as input, and the texture maps are generated by fine-tuning a stable diffusion model. This method achieves a better CLIP score and faster inference compared with prior art.

**Strengths:**

- The paper is well-written and easy to follow.

- The proposed method is technically sound. Most design choices are well-motivated.

**Weaknesses:**

- The baseline DreamFace seems to generate higher-quality results. Also, DreamFace considers 4K resolution texture maps while the proposed method only considers 256x256.

- All generations have the same skin color, e.g. Mark Zuckerberg and Morgen Freeman in Figure 8.

This paper’s results are qualitatively not as impressive as its baseline DreamFace,  with significantly lower resolution and skin color variation. I also have concerns regarding the fact that the training data selection step.

**Questions:**

Overall, this paper’s results are qualitatively not as impressive as its baseline DreamFace, with significantly lower resolution and skin color variation. I also have concerns regarding the fact that the training data selection step. My questions include:

- Can the proposed method be used for 4K generation or is there any fundamental limitation?
- Why do the generations have very limited skin color variations?
- The authors mention that they use SDS optimization to obtain 600 samples while selecting only 50 samples for training to “ensure a balance of gender, ethnicity, and age”. Why selection is needed, and why not just balancing the input text prompts? Also, 50 training samples sound very limited. Why not use more samples?
- How does the proposed method compare with the SDS optimization pipeline which is used for training data generation?

---

> ### Author Response · Authors · 2023-11-18
> **Response to Reviewer 3kTT**
>
> Thank you for your valueable comments. Please refer to the general response for information about 4K resolution results. Here are the answers to the other questions:
>
> __Q1: skin color__
>
> __A1__: Besides texture resolution, reviewers are also concerned about the skin color generated by our method.
> This issue is partly caused by the lighting settings in rendering software, such as Blender. We have updated the relevant images in the main paper and the appendix to better highlight the skin color variations.
>
> __Q2: Why selection of training examples is needed__
>
> __A2__: We use only 50 samples because, generally, LoRA training requires just a few dozen samples. Besides, due to the inductive bias of SD, the generated data by SDS is not very fair in terms of gender, ethnicity, and age. Therefore we perform a simple selection from 600 sample to form our training set. Moreover, we empirically find that using more sample for LoRA finetuning not only leads to longer finetuning time, but also results in overfitting problem. The finetuned model will concentrate on generating normal-looking humans, while having poor generation of contents that are not part of the dataset and require large areas of color blocks, such as The Joker and Batman.
>
>
> __Q3: Comparison with SDS__
>
> __A3__: We have updated the results in our appendix. As shown in Fig.10 in the appendix, the data generated by SDS does contain some blurry samples, but the final generated results are clearer, thus showing the effectiveness and necessity of our method.

---

> > ### Comment · Reviewer_3kTT · 2023-11-22
> > **Official Comment by Reviewer 3kTT**
> >
> > Thank authors for their efforts in addressing my concerns. However, the updated 4K textures do not seem to capture high-resolution details, e.g. in Figure 3, the proposed method generates less details than DreamFace. Regarding skin color variation, although the authors provide generations with other skin colors, these colors seem to be 4 discrete options (in Figure 8 and 9) lacking fine-grained variations.

---

### Official Review · Reviewer_ggv2 · 2023-10-31

**Soundness:** 2 fair
**Presentation:** 2 fair
**Contribution:** 2 fair
**Rating:** 5
**Confidence:** 4

**Summary:**

This paper proposes a novel approach for text-guided 3D animatable head avatar generation where a 3D avatar with desired facial characteristics is generated based on input textual prompts. It draws inspiration from recent works on diffusion-based text-to-3D approaches such as DreamFusion. The authors propose learning shape parameters of a FLAME-based 3D head model using a pretrained CLIP text encoder. A pretrained Latent Diffusion model is fine-tuned using an additional mean-texture token for generalized learning of the facial texture. The proposed method adopts SDS technique to generate training data for training the shape and texture generator. The main contribution is the reduction of inference time complexity for 3D avatar generation. The proposed method also does not require 3D annotated data for training. Qualitative and quantitative comparison results are presented with state-of-the-art methods on text-to-3D methods.

**Strengths:**

1.	The proposed method claims the lowest test time complexity among existing test-to-3D models that can generate 3D faces. There is substantial reduction in inference time (1 min) compared to the most efficient method DreamFace, which takes around 5 mins for the optimization. However the texture resolution is lower than DreamFace.
2.	Qualitative results denote decent quality of 3d faces.  The reconstructed avatar of celebrity faces are bearing resemblance to the real people.
3.	The method supports specific prompts for tasks such as generating special characters (animation) and editing shape and style.

**Weaknesses:**

•	Novelty and Significance of contributions:
The novelty of the proposed method appears slightly limited. Similar to DreamFace, pretrained CLIP and LDM models are used in the avatar generation with independent geometry(shape) and texture generators. The idea of using a mean texture token is novel, but similar to ideas have been explored in the form of pre-defined identity token in DreamBooth, and domain-specific prompt tuning in DreamFace, Introduction point 2 mentions challenges in animation due to implicit representations as limitation of existing text-to-3D methods. However DreamFace uses the ICT-FaceKit face model that can be integrated with existing animation pipelines, so the benefit obtained from using FLAME model in the current work is not clear. DreamFace also additional benefits of hair selection and video-driven animation generation. It is not evident from the paper how much the state-of-the-art in text-guided-3D face avatar generation will be advanced by the proposed method. Although the paper claims reduction in inference time, there are doubts about the generalization ability of the method in generating arbitrary high-fidelity avatars of varying age, skin colours etc, given that the training data consists of manually selected 50 training samples generated by SDS optimization
Writing Issues:
•	Unclear writing:
o	(Page 2) “need for a cumbersome two-stage generation process” – what two-stage generation process.
o	(Page 2) “meticulously crafted to encapsulate essential human head texture information.” - How
o	(Page 3) “we further propose other specific design to generate high-quality animatable 3D head avatars” – what designs.
o	“the renewed text prompts can contribute to fine-grained personalized characteristics with high fidelity of identity”- doesn’t make sense
o	“common texture-wise features shared by human beings.”
o	Epsilon is not defined in Equation 1.
 •	Typos:
	“our propose generalized shape”  in Page 5
         Equation 2 \phi() needs to be replaced by e_\phi()

•	Missing citations :
o	“Existing methodologies [??] typically leverage SDS”
o	 “remarkable strides achieved in diffusion-based text-to-3D models [??]”
o	“While these [??] SDS-based approaches”
o	“Leveraging readily available off the-shelf models [??]”
o	“[Articulated Diffusion]”

Experimental Results:
•	3D view (other than frontal) should have been included similar to existing works such as DreamFace. In the absence of a supplementary video it is hard to assess the qualitative results.
•	User Study needed to assess the perceptual quality of the generated results.
•	More detailed ablation study should be presented, the significance of the mean texture token should be justified using quantitative metrics.
•	Some failure cases should be present to illustrate limitations of the method.

**Questions:**

1.	The description “data-free” strategy appears ambiguous as it also mentioned that SDS is used to generate training data. More clarity is needed on the training strategy in Section 3.3. Is a pretrained stable diffusion model being finetuned for the training data preparation?  What kind of candidate text prompts are used for the geometry and UV texture generation (few examples) How is it ensured that the “training data” generated using SDS sufficiently accurate for generalized performance at inference time.
2.	Which pre-trained LDM models are used for finetuning?
3.	The significance of the mean-texture token is not clear from the results. How is the mean-texture token prompt obtained at test time?
4.	Is the mean-texture token sufficient to finetune a pretrained LDM (trained on diverse images) to the specific task of the Face UV texture generation. How is it ensured that the generated texture is consistent with face geometry? In the absence of UV texture ground truth to finetune pretrained LDM for texture how is the accuracy ensured at inference time?
5.	The paper mentions “we set this parameter to a relatively low value and obtain more realistic, real-life outcomes.” Is there any ablation done on the guidance scale parameter to justify this statement?

---

> ### Author Response · Authors · 2023-11-18
> **Response to Reviewer ggv2**
>
> We thank reviewer ggv2 for the valuable time and constructive feedback.
>
> __Q1: Novelty in terms of pretrained CLIP and LDM__
>
> __A1__: Thank you. We would like to stress that using pretrained foundation models like CLIP and LDM is common in the current vision tasks due to their strong capability. We have not listed such techniques as our contribution. Instead, as mentioned in the general response, our novel pipeline for 3D head avatar generation along with the data-free training strategy and the usage of mean texture token for representing general human facial details are the key contribuion of our paper.
>
> __Q2: Mean texture token v.s. DreamBooth and domain-specific prompt tuning__
>
> __A2__: We have explained the difference between our mean texture token and identifier in DreamBooth in Sec.3.2. We quote the original content here: "It is noteworthy that while the proposed mean-texture token is similar to the unique subject identifier utilized in DreamBooth, the actual effect is indeed different. The identifier in DreamBooth is aimed to represent a specific subject. In contrast, our mean-texture token helps complement text prompts by representing common texture-wise features shared by human beings." Moreover, our method differs from DreamBooth in implementation. DreamBooth finetunes the entire large model and requires a prior preservation loss. In contrast, our approach employs the LoRA method for training, which adds low-rank layers to the freeze original Stable Diffusion, reducing overfitting and training consumption due to fewer parameters.
>
> As for the domain-specific prompt tuning in DreamFace, we think it has essential difference with our mean texture token. The domain-specific prompt tuning was proposed to help model identify the unwanted domain data resulted from the collection. In short, the method is used for data filtering. On the contrary, our mean texture token is proposed to represent the prior knowledge of human face textures, as mentioned in Sec.3.2.
>
> __Q3: ICT-FaceKit v.s. FLAME__
>
> __A3__: Thank you. The advantage of FLAME introduced in our paper is in comparison to latent representation methods like Nerf and DMTet. With this linear shape parametric model, we can easily control the shape using 100-dimensional parameters and easily integrate with a CG pipeline to achieve downstream tasks, such as video-driven animation.
>
> Both our FLAME model and DreamFace's ICT-FaceKit belong to the 3DMM parametric model category and share the advantages mentioned above. The differences lie in:
>
> (1) The texture coverage of ICT-FaceKit does not include the entire face, covering only the ears and the frontal face without eyes, while our FLAME model's texture covers the entire head, which can be seen in the videos in our supplementary material.
>
> (2) Our usage of the 3DMM model also differs from DreamFace. DreamFace samples shape parameters from a multivariate normal distribution N(0,1) to obtain one million candidates, followed by 300 carving steps. In contrast, we design the AniHead model pipeline to directly learn shape parameters, inferring the shape parameter vector directly from the prompt. This not only results in higher shape-text matching (as shown in Tab.1 and Fig.3) but also reduces inference time (ours can generate shape within seconds).
>
> __Q4: more generation results demonstrating varying age, skin colors__
>
> __A4__: As both our method and DreamFace use parametric models as shape model, they can share a CG pipeline to achieve effects such as hair selection and video-driven animation, as demonstrated in the gif in Fig.5. We have updated Fig.4 in the main paper to demonstrate the editing capabilities for age and skin colors. As can be seen from the results, our model is versatile to multiple types of editing on different people.
>
> __Q5: Unclear writing and missing citations__
>
> __A5__: Thanks for pointing out. We have updated these problems in the new version.
>
> __Q6: 3D view__
>
> __A6__: We have provided the required videos in the supplementary material. The results are consistent with the original results in our paper, supporting the efficacy of our method.
>
> __Q7: data-free training strategy__
>
> __A7__: Thank you. We would like to kindly point out that 'data-free' refers to the fact that manual data collection is not required. In the SDS generation process, the only preparation needed is providing a prompt, such as "a portrait of {name} with neck and shoulder, without hair", which can be easily created. Moreover, the generalization ability has been shown in results in Fig.4. Given limited training data generated by SDS, these results further prove the effectiveness of our proposed pipeline.
>
> We would like to clarify that the Stable Diffusion model is freeze during the SDS optimization process, because SDS only needs to leverage the prior information from Stable Diffusion.

---

> > ### Author Response · Authors · 2023-11-18
> >
> > __Q8: which pretrained LDM is used?__
> >
> > __A8__: We use the pretrained LDM checkpoint stable-diffusion-2-1-base from stabilityai, and we found that stable-diffusion-v1-5 from runwayml also works well. Therefore, the choice of LDM base model checkpoint does not significantly impact the performance of our method.
> >
> >
> > __Q9: mean texture token__
> >
> > __A9__: We would like to clarify that the mean-texture token is not a trainable parameter in our work. It is represented by a rare token, specifically 'shs'. During inference, simply adding this token to the beginning of the text input is enough for the model to understand the corresponding semantic meaning. As for the effectiveness of the mean texture token, we have showcased in Fig.6 that this unlearnable token can better represent human facial texture than natural language.
> > We also demonstrate in Fig.11 in the appendix that, with the mean texture token, the UV geometry of delicate parts may be misaligned, such as the nose and eye regions shown in Fig.11. In contrast, our design leads to better UV map positioning. As we have analyzed, the mean texture token strengthens the UV relation map.
> >
> > __Q10: how is the accuracy ensured at inference time?__
> >
> > __A10__: Indeed, in our training process, we do not use ground truth datasets. Instead, we treat the optimization results from SDS optimization as ground truth, given the strong capabilities of SDS. Moreover, this approach allows us to achieve data-free training. Our experimental results in Fig.10 demonstrate that even if the training data contains some blurry examples, it does not negatively impact the final generated results. The generated images exhibit clear skin textures and patterns.
> > Whereas some of the SDS generated data are blurry in details, the generated results are clear and can easily be generated to out-of-domain examples such as 'The Joker'.
> >
> > __Q11: the effect of CFG value__
> >
> > __A11__: We have updated Fig.7 in our appendix to demonstrate that relatively low CFG value (7.5) leads to more realistic, real-life outcomes.

---

> > > ### Comment · Reviewer_ggv2 · 2023-11-22
> > >
> > > Thanks for the response to the questions. It seems that the qualitative results in Fig. 3 in the revised submission have been updated with 4K texture. It appears that the results reported in the original submission had better texture quality in the original resolution. The 4K texture has noticeable artifacts in (e.g mouth) in Figs. 3 and 4.

---

### Official Review · Reviewer_9Wmk · 2023-11-01

**Soundness:** 2 fair
**Presentation:** 2 fair
**Contribution:** 2 fair
**Rating:** 6
**Confidence:** 3

**Summary:**

In this paper, the authors introduce a comprehensive pipeline for the generation of 3D heads. Their approach begins with the application of a Score Distillation Sampling (SDS) technique to create training data for FLAME-based models. Subsequently, they employ this paired dataset to train generators for both shape and texture. To evaluate the efficacy and efficiency of their method compared to baseline techniques, the authors conducted a series of experiments, the results of which are presented in the paper.

**Strengths:**

This innovative method presents a unique pipeline for text-to-3D head generation that distinguishes itself in several ways. Notably, it does not rely on annotated datasets for training, making it exceptionally versatile. Additionally, the utilization of FLAME as the 3D representation in this method contributes to faster inference times, setting it apart from other baseline approaches.

**Weaknesses:**

My apprehension revolves around the generative quality constrained by the use of FLAME. It appears that the resulting shape and texture may fall short of the realism achieved by DMTet-based or Nerf-based methods. Moreover, there seems to be a limitation in the ability to synthesize 3D hair components.

Furthermore, it's worth noting that the methods employed in this approach draw heavily from existing techniques. For instance, the process of generating the training dataset bears a resemblance to DreamFusion, albeit with the incorporation of the FLAME representation.

**Questions:**

I'd like to pose two questions:

In Figure 4, I'm curious about how the model manages to synthesize 3D hair for "Taylor Swift." It seems like a noteworthy achievement, and I'm interested in understanding the underlying techniques.

In Figure 2, during the training data preparation stage, there appears to be a differentiation in the input for Stable diffusion, involving both shader images and textured images. I'd like clarification on the purpose of these distinct inputs for various steps and how they relate to the rendering equation and the overall model.

---

> ### Author Response · Authors · 2023-11-18
> **Response to Reviewer 9Wmk**
>
> We thank reviewer 9Wmk for the valuable time and constructive feedback.
>
> __Q1: synthesize 3D hair for "Taylor Swift."__
>
> __A1__: Thank you for your comment. As mentioned in Sec 3.2, the addition of accessories such as hats and hair are solved by post-processing steps. This procedure is similar to DreamFace, where candidate hair is selected and added to the generated avatars. Specifically, we put the generated 3D head into Blender and add a default hair accessory for it. We have updated the caption of Fig.1 for better understanding.
>
> __Q2: difference in the input for SD during data preparation__
>
> __A2__: We utilize SDS to generate both shape and texture training data. Originally, SDS optimizes latent 3D parameters like NeRF for generation. Since we want to generate decoupled shape and texture instead of single 3D shape, we have to optimize two sets of parameters controling shape and texture respectively. Specifically, shape paramter $S\in R^{1 \times 100}$
>  is optimized with SDS. $S$ can be mapped to 3D head shape by fixing the pose and expression parameters to 0 in the FLAME model. Formally, $
>  T, M_{normal} = FLAME(S)$, where $T$ is mesh shape and $M_{normal}$ is the according normal map.
>  Then, we use a differentiable renderer $R$ with a random camera pose $C$ to obtain the shader image $I = R(T, C, M_{normal})$. We then encode $I$ into the latent space to obtain $z_t$, and we use the SDS loss to get the optimized shape $S'$.
>
> For texture, we fix the shape parameters $\hat{S} = S'$ and set the parameter to be optimized as texture map $M_{\text{tex}}$, which is initialized as the mean texture for four different skin colors. Similarly, we obtain $I = R(T, C, M_{\text{tex}})$, which reflects the effect of mapping the texture map to be optimized onto the 3D shape.
>
> We have updated this detail in the appendix.
>
>
> __Q3: quality constrained by the use of FLAME__
>
> __A3__: It is true that FLAME may have limitations in representing certain accessories such as hair or hats. However, using a parametric model has its advantages:
>
> (1) The parametric model incorporates a human head prior, providing constraints that enable the generation of more reasonable results, while implicit representation methods struggle or fail to generate reasonable results (as shown in Fig.3). Both shape and texture are more controlable and reasonable, which had also mentioned in our overall respose.
>
> (2) Our approach is animatable, allowing for easy integration with off-the-shelf driving methods to create animations, as demonstrated in the gif in Fig.5. In contrast, implicit methods, although capable of representing hair, are not animatable and lack downstream applications.
>
> (3) For static 3D head avatars, adding hairs by post-processing, as adopted by us and DreamFace, would not lead to significant performance gap with latent-based methods.
>
> We have updated the discussion in our appendix.

---

> > ### Comment · Reviewer_9Wmk · 2023-11-22
> >
> > Thank you for your feedback.

---

### Author Response · Authors · 2023-11-18
**General response**

Dear Chairs and all Reviewers,

We would like to thank all the reviewers for their detailed and thoughtful comments. We are particularly encouraged that all the reviewers found our proposed AniHead to be innovative and versatile (9Wmk) with decent 3d face quality (ggv2). Moreover, they commended our paper to be a well-written and solid one (3kTT). We find that some reviewers have misunderstood certain modules in our paper, and we have clarified these points accordingly. Among all comments, we would like to highlight some of the general questions from the reviewers, including novelty of our method, quality in terms of texture resolution and skin colors.

* __novelty__

As mentioned in Sec.1, our contribution stems from two aspects.

(1) We propose a generalized 3D head avatar generation framework, which enjoys significantly faster inference speed than previous methods, leading to better practical usage. By utilizing the head prior provided by the FLAME model, our method is capable of generating reasonable shape results with fine-grained and plausible control over the shape, while implicit representation approaches such as DMTet-based or Nerf-based methods often struggle or fail to do so. As a result, our method produces better skin texture compared to other approaches and does not suffer from overly smooth colors.

(2) We innovatively explore the potential of SDS for 3D avatar training data collection instead of direct inference. Such a technique helps our method achieve good performance without requiring large-scale training data, which can further benefit the future research in this area.

* __texture resolution__

As mentioned by Reviewer ggv2 and 3kTT, the resolution of results in our initial submission are much lower than that of DreamFace. We would like to kindly point out that __high texture resolution of DreamFace stems from off-the-shelf super-resolution pipeline__. Specifically, DreamFace first generates 512x512 textures, then utilizes RestoreFormer [1] for face restoration and Real-ESRGAN [2] for superresolution from 512x512 to 4K. This means no matter what resolution our method can generate, it is still possible for us to produce 4K results. To prove this, we adopt a similar pipeline, in which the 256x256 textures are first restored to 1024x1024 using RestoreFormer++ [3], then super-resolution are utilized to generate 4K textures. The corresponding results are shown in Fig.4, in which we can find that the generation results are further refined after super-resolution. Moreover, by merging this pipeline with our AniHead, the inference time for each text prompt is still about 1 minute, which further shows the essential efficacy of our method.



* __comparison with DreamFace__

Our main difference with DreamFace lies in the fact that their method relies on scanned data, which is generally not feasible for most organizations. In contrast, our approach uses SDS optimization to obtain 50 samples, which are then used to finetune the large model. This makes our method more practical in terms of resource consumption and data requirements.
Here is a comparison of our method and DreamFace:
|   | DreamFace | Ours |
|---|-----------|------|
|Dataset|38400 registered mesh, 614400 images|50 shape&texture|
|Training consumption|12 hours, 2 A1000 GPUS|0.5 hour, 1V100 GPU|
|Inference Time|~5 mins|~1 min|
|Shape Model|choose from one million pre-defined shapes + 300 steps of detail carving|directly infer 100-dimension shape parameters|
|CLIP score(Text matching)|0.2934|0.3161|


[1] Wang, Zhouxia, et al. "Restoreformer: High-quality blind face restoration from undegraded key-value pairs." Proceedings of the IEEE/CVF Conference on Computer Vision and Pattern Recognition. 2022.

[2] Wang, Xintao, et al. "Real-esrgan: Training real-world blind super-resolution with pure synthetic data." ICCV 2021.

[3] Wang, Zhouxia, et al. "RestoreFormer++: Towards Real-World Blind Face Restoration From Undegraded Key-Value Pairs." TPAMI 2023.

---

> ### Author Response · Authors · 2023-11-18
> **Summary of Revisions**
>
> The major changes are as follows:
>
> 1. We've updated Fig.1-3 in the main paper and Fig.8-9 in the appendix with more suitable material and lighting settings, to better show the skin color variety as suggested by 3kTT, and with 4K textures, as mentioned by ggv2 and 3kTT.
>
> 2. We've added more editing results in Fig.4 to demonstrate the editing capabilities for age and skin colors, as suggested by ggv2.
>
> 3. We've added SDS Training Details and in appendix as suggested by 9Wmk.
>
> 4. We've further discussed our choice of FLAME model in appendix according by 9Wmk.
>
> 5. We've updated the caption of Fig.1 for better understanding of the CG pipeline postprocessing effect, according by 9Wmk.
>
> 6. We've added Fig.11 in appendix to show the effect of mean texture token, as suggested by ggv2.
>
> 7. We've corrected the unclear writing and missing citations as pointed out by ggv2.
>
> 8. We've provided 3D view videos in the supplementary material, as suggested by ggv2.
>
> 9. We've added Fig.7 in appendix to better demonstrate our Classifier-free guidance discussion, as suggested by ggv2.
>
> 10. We've added Fig.10 in appendix to compare the result from SDS optimization and inference, as suggested by 3kTT.

---

### Author Response · Authors · 2023-11-21

Dear Reviewers,

We are deeply grateful for the time and energy you have invested in evaluating our work. We acknowledge that you may have a demanding schedule. As we approach the end of the authors-reviewer discussion phase, we would like to gently draw your attention to our responses. Our goal is to understand if our replies have adequately addressed your queries and to determine if there are any further questions or aspects you wish to explore.

We eagerly anticipate the chance for more dialogue with you. Thank you for your careful and considerate review.

Best regards,

The Authors

---

### Meta-Review · Area_Chair_1h63 · 2023-12-05

**Metareview:**

This paper aims at the task of text-to-3D portrait. It proposes a data-free training schema with two stages. In the first stage, a set of samples are synthesized by combined parametric model (FLAME) and SDS. In the second stage, two feed-forward networks for shape and texture are learned using the synthesized data. Reviewers agree that the paper is well-written, proposing a data- and time-efficient method for the chosen task. However, reviewers also raised questions regarding unrealistic results (e.g., limitation in modeling hair, skin color), limited novelty, inferior performance (especially in higher resolution).

**Justification For Why Not Higher Score:**

The major weakness of this paper is novelty and performance, as pointed by multiple reviewers simultaneously. The novelty of the paper is limited since most of its components are proposed by existing works. The method also cannot generate realistic hair, skin color, especially in higher resolution. Based on the discussions between reviewers and authors, I think the paper does not meet acceptation standard and thus recommend for rejection.

**Justification For Why Not Lower Score:**

N/A

---

### Decision · Program_Chairs · 2024-01-16

Reject